# Non-robust Features through the Lens of Universal Perturbations

## Abstract

Recent work ties adversarial examples to existence of non-robust features: features which are susceptible to small perturbations and believed to be unintelligible to humans, but still useful for prediction. We study universal adversarial perturbations and demonstrate that the above picture is more nuanced. Specifically, even though universal perturbations—similarly to standard adversarial perturbations—do leverage non-robust features, these features tend to be fundamentally different from the "standard" ones and, in particular, non-trivially human-aligned. Namely, universal perturbations have more human-aligned locality and spatial invariance properties. However, we also show that these human-aligned non-robust features have much less predictive signal than general non-robust features. Our findings thus take a step towards improving our understanding of these previously unintelligible features.

## 1 Introduction

Modern deep neural networks perform extremely well across many prediction tasks, but they largely remain vulnerable to adversarial examples (Szegedy et al., 2014). Models' brittleness to these small, imperceptible perturbations highlights one alarming way in which models deviate from humans. Recent work gives evidence that this deviation is due to the presence of useful *non-robust* features in our datasets (Ilyas et al., 2019). These are brittle features that are sensitive to small perturbations—too small to be noticeable to humans, yet capture enough predictive signal to generalize well on the underlying classification task. When models rely on non-robust features, they become vulnerable to adversarial examples, as even small perturbations can flip the features' signal.

While prior work gives evidence of non-robust features in natural datasets, we lack a more fine-grained understanding of their properties. In general, we do not understand well how models make decisions, so it is unclear how much we can understand about these features that are believed to be imperceptible. A number of works suggest that these features may exploit certain properties of the dataset that are misaligned with human perception (Ilyas et al., 2019), such as high-frequency information (Yin et al., 2019), but much remains unknown.

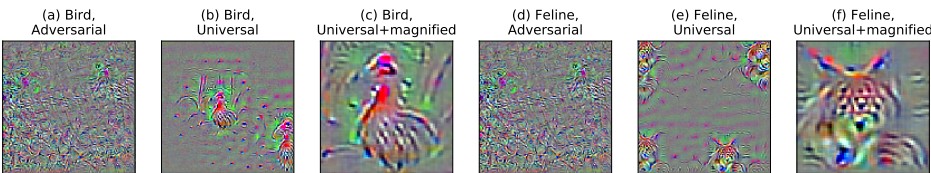

Figure 1: $\ell_2$ adversarial perturbations ($\epsilon = 6.0$) for two target classes, bird and feline, on ImageNet-M10: (a,d) standard adversarial perturbations on a single image, (b,e) universal adversarial perturbations, and (c,f) zooming in on the most semantic patch.

In this work, we illustrate how we can isolate more human-aligned non-robust features by imposing additional constraints on adversarial perturbations. In particular, we revisit universal adversarial perturbations (Moosavi-Dezfooli et al., 2017a), i.e. adversarial perturbations that generalize across many inputs. Prior works have observed that these perturbations appear to be semantic (Hayes & Danezis, 2019; Khrulkov & Oseledets, 2018; Liu et al., 2019). We demonstrate that universal perturbations

possess additional human-aligned properties different from standard adversarial perturbations, and analyze the non-robust features leveraged by these perturbations. Concretely, our findings are:

**Universal perturbations have more human-aligned properties.** We show that universality adversarial perturbations have additional human-aligned properties that distinguish them from standard adversarial perturbations (e.g., Figure 1). In particular, (1) the most semantically identifiable local patches inside universal perturbations also contain the most signal; and (2) universal perturbations are approximately *spatially invariant*, in that they are still effective after translations.

**Non-robust features can be semantically meaningful.** We show that universal perturbations are primarily relying on non-robust features rather than robust ones. Specifically, we compare the sensitivity of natural and (adversarially) robust models to rescalings of these perturbations to demonstrate that universal perturbations likely rely on non-robust features. Together with our first finding, this shows that some non-robust features can be human-aligned.

**Universal perturbations contain less non-robust signal.** We find that the non-robust features leveraged by universal perturbations have less predictive signal than those leveraged by standard adversarial perturbations, despite being more human-aligned. We measure both (1) generalizability to the original test set and (2) transferability of perturbations across independent models, following the methodology of Ilyas et al. (2019). Under these metrics, universal perturbations consistently obtain non-trivial but substantially worse performance than standard adversarial perturbations.

## 2 PRELIMINARIES

We consider a standard classification task: given input-label samples $(x, y) \in \mathcal{X} \times \mathcal{Y}$ from a data distribution $\mathcal{D}$, the goal is to to learn a classifier $C : \mathcal{X} \to \mathcal{Y}$ that generalizes to new data.

**Non-robust vs. robust features.** Following Ilyas et al. (2019), we introduce the following terminology. A *useful* feature for classification is a function that is (positively) correlated with the correct label in expectation. A feature is *robustly useful* if, even under adversarial perturbations (within a specified set of valid perturbations $\Delta$) the feature is still useful. Finally, a *useful, non-robust* feature is a feature that is useful but not robustly useful. These features are useful for classification in the standard setting, but can hurt accuracy in the adversarial setting (since their correlation with the label can be reversed). For conciseness, throughout this paper we will refer to such features simply as *non-robust features*.

**Universal perturbations.** A *universal adversarial perturbation* (or just *universal perturbation* for short), as introduced in Moosavi-Dezfooli et al. (2017a), is a perturbation $\delta$ that causes the classifier $C$ to predict the wrong label on a large fraction of inputs from $\mathcal{D}$. We focus on *targeted* universal perturbations that fool $C$ into predicting a specific (usually incorrect) target label $t$. Thus, a (targeted) universal perturbation $\delta \in \Delta$ satisfies $P_{(x,y) \sim \mathcal{D}}[C(x + \delta) = t] = \rho$, where $\rho$ is the *attack success rate* (ASR) of the universal perturbation. The only technical difference between universal perturbations and (standard) adversarial perturbations is the use of a single perturbation vector $\delta$ that is applied to all inputs. Thus, one can think of universality as a constraint for the perturbation $\delta$ to be input-independent. Universal perturbations can leverage non-robust features in data, as we show here; we refer to these simply as *universal non-robust features*.

$\ell_p$ **perturbations.** We study the case where $\Delta$ is the set of $\ell_p$-bounded perturbations, i.e. $\Delta = \{\delta \in \mathbb{R}^d \mid ||\delta||_p \leq \epsilon\}$ for $p = 2, \infty$. This is the most widely studied setting for research on adversarial examples and has proven to be an effective benchmark (Carlini et al., 2019). Additionally, $\ell_p$-robustness appears to be aligned to a certain degree with the human visual system (Tsipras et al., 2019).

### 2.1 COMPUTING UNIVERSAL PERTURBATIONS

We compute universal perturbations by using projected gradient descent (PGD) on the following optimization problem:

$$\min_{\delta \in \Delta} \mathbb{E}_{(x,y) \sim \mathcal{D}}\left[\mathcal{L}(f(x + \delta), t)\right] \tag{1}$$

where $\mathcal{L}$ is the standard cross-entropy loss function for classification, and $f$ are the logits prior to classification. and $t$ is the target label. While many different algorithms have been developed for

computing universal perturbations with varying attack success rates, we do not consider them as achieving the highest rate is not necessary for our investigations. As observed in Moosavi-Dezfooli et al. (2017a), universal perturbations trained on only a fraction of the dataset still generalize to the rest of the dataset, so in practice it suffices to approximate the expectation in equation 1 using a relatively small batch of data drawn from $\mathcal{D}$. We call the batch we optimize over the *base set*, and use $K$ to refer to its size. We describe the selection of all hyperparameters in Appendix B.

## 2.2 EXPERIMENTAL SETTINGS

**Datasets.** We conduct our experiments on a subset of ImageNet (Deng et al., 2009) and CIFAR-10 (Krizhevsky et al., 2009). The Mixed10 ImageNet dataset (Engstrom et al., 2019), which we refer to as ImageNet-M10, is formed by sub-selecting and grouping together semantically similar classes from ImageNet; it provides a more computationally efficient alternative to the full dataset, while retaining some of ImageNet's complexity (see Appendix B.1 for more details about the dataset). In the main text, we focus our results on ImageNet-M10, and in the Appendices we provide additional results on CIFAR-10 as well as visualizations from a full ImageNet model.

**Models.** We use the standard ResNet-18 architecture (He et al., 2016) unless mentioned otherwise.

## 3 PROPERTIES OF UNIVERSAL PERTURBATIONS

We find that universal perturbations, in contrast to adversarial perturbations, are much more aligned with human perception. Beyond visualizations, we demonstrate this in the following ways: (1) local patches that are most semantically identifiable with the target class also contain the most predictive signal; and (2) universal perturbations are approximately spatially invariant, whereas standard perturbations are not.

### 3.1 PERCEPTUAL ALIGNMENT

**Visual comparison with standard adversarial perturbations.** Standard $\ell_p$ adversarial perturbations are often thought to be incomprehensible to humans (Szegedy et al., 2014; Ilyas et al., 2019). Even when magnified for visualization, these perturbations are not identifiable with their target class. In contrast, universal perturbations are visually more interpretable: when amplified, they contain local regions that we can identify with the target class (see Figure 2 for example universal perturbations, and Figure 1 for a comparison to standard perturbations). Prior works have also observed that universal perturbations resemble their target class (Hayes & Danezis, 2019) and contain meaningful texture patterns (Khrulkov & Oseledets, 2018; Liu et al., 2019). Next, we go beyond simple visualizations by identifying two additional human-aligned properties of universal perturbations.

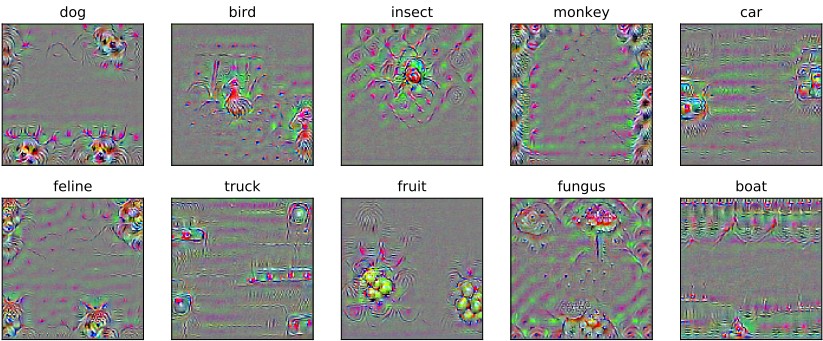

Figure 2: A sample of $\ell_2$ universal perturbations ($\epsilon = 6.0$) for all target classes on ImageNet-M10: we observe patterns resembling dog and cat (feline) faces, birds, outlines of cars and trucks, an insect, and mushrooms (fungus).

**Predictive signals are localized to most semantically identifiable patches.** While universal perturbations contain parts that are semantically identifiable with the target class, they also contain less

identifiable regions; a priori, it is unclear which parts influencing the model. To investigate this, we *randomly* select different local patches of the perturbation, evaluate their ASR in isolation, and inspect them visually. For $\ell_2$ perturbations, the patches vary widely in norm, so a possible concern is that the model is only reacting to the patches with the highest norm; to correct for this, we linearly scale up all patches to have the same norm as the largest norm patch. Even after normalization, the patches with the highest ASR are more semantically identifiable with the target class (Figure 3). For $\ell_\infty$ perturbations, where all patches have similar norms (so no further normalization is done), we also observe that their ASR is proportional to their visual saliency. This demonstrates that the model is indeed reacting primarily to the most salient parts of the perturbation.

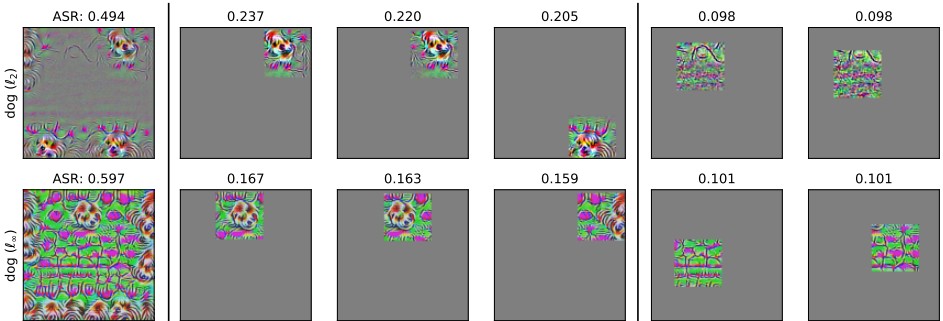

Figure 3: Local analysis of universal perturbations on ImageNet-M10: 64 random $80\times80$ patches are isolated and evaluated for their ASR. First column shows the original perturbation for the target class (top: $\ell_2$; bottom: $\ell_\infty$); next five columns show three patches with the highest ASR, and two with the lowest. Top of each perturbation indicates its ASR on the test set. Patches are normalized.

## 3.2 SPATIAL INVARIANCE

Adversarial perturbations are not only unintelligible but also extremely brittle to translations, as we demonstrate. In contrast, we find that universal perturbations are translationally invariant to a large degree. As spatial invariance is one of the key properties of the human visual system (Hubel & Wiesel, 1968), this illustrates another way that universal perturbations are more human-aligned than standard perturbations.

We quantify spatial invariance by measuring the ASR of translated perturbations. Specifically, we compute targeted standard adversarial perturbations on a sample of 256 images from the ImageNet-M10 test set. Then, we evaluate the ASR of different translated copies of these perturbations over the test set (copies of each perturbation are evaluated only on its corresponding image). For comparison, we take a precomputed set of universal perturbations of the same norm, and evaluate them across all images. Figure 16 shows that universal perturbations still achieve non-trivial ASR after translations of varying magnitudes. In contrast, standard adversarial perturbations achieve a chance-level $10\%$ ASR when shifted by more than eight pixels (two grid cells in Figure 16). This illustrates that the model reacts differently to universal perturbations than to standard adversarial perturbations.

## 4 UNIVERSAL PERTURBATIONS LEVERAGE NON-ROBUST FEATURES

The predominant belief about non-robust features is that they are incomprehensible to humans. We saw, however, that universal perturbations have human-aligned properties. And since universal perturbations are bounded in norm, they likely rely on non-robust features to fool the model. We might then conclude that there exist non-robust features that are human-aligned.

However, there is a possibility of *robust feature leakage* (Goh, 2019): while a small perturbation cannot entirely flip the correlation of a robust feature, it can still induce a small correlation by perturbing the feature. If universal perturbations primarily rely on robust features, we cannot necessarily conclude that the nice properties we observed earlier are tied to non-robust features. In

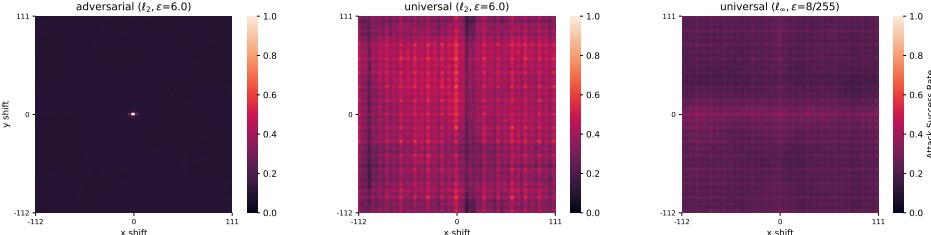

Figure 4: Evaluation of translated adversarial and universal $\ell_2, \ell_\infty$ perturbations for the target class `bird` on ImageNet-M10. We evaluate a subsampled grid with strides of four pixels. The value at coordinate $(i, j)$ represents the average ASR when the perturbations are shifted by $i$ pixels right and $j$ pixels up, with wraparound to preserve information; the center pixel at $(0, 0)$ represents the ASR of the original unshifted perturbations. For universal perturbations, the ASR at each location is averaged over ten perturbations.

this section, we provide evidence that robust leakage is unlikely to be a primary contribution to the signal.[1]

**Scaling analysis on natural vs. robust model** Our analysis based on the following premise: if the universal perturbations are primarily relying on small perturbations of robust features, then a robust model should eventually react when we amplify the signal in the universal perturbations. Below, we first show that simply linearly scaling the perturbation can effectively tune the strength of the signal. More precisely, given a universal perturbation $\delta$, we evaluate the ASR of $t \cdot \delta$, where $t$ is a scaling parameter. [2] When we perform this scaling on a natural (non-robust) model, the ASR increases as we increase $t$ (Figure 5). (This holds even if networks used to train the universal perturbation and the one used to evaluate its ASR differ.) If there was significant robust leakage, we would expect a similar qualitative behavior for the robust model. Instead, we find that the robust model does *not* react to any scaling of the perturbation. On the other hand, a universal perturbation of norm $\epsilon = 30.0$[3] is computed on an adversarially-trained robust model ($\epsilon = 6.0$) still fools the natural model, even when scaled down. Together, these findings suggest that universal perturbations computed on the natural model are unlikely to leverage robust features used by the robust model.

In fact, a more fine-grained scaling analysis shows that the signal is indeed primarily amplified in the semantic parts of the image. In Section 3 we already observed that models are most sensitive to the most semantic parts of universal perturbations; here we check that the order remains consistent across various scales. We repeat the above scaling evaluation on natural vs. robust models, except we scale only patches of the perturbation, of varying levels of semantic content (Figure 5b). As before, we find that none of the patches, even after scaling, trigger the robust network. On the non-robust models, the relative effectiveness of local patches is consistent with their semantic content: patches that are more semantically identifiable have more signal at all scales.

The above analysis shows that universal perturbations are likely primarily targeting non-robust features. Combined with our findings earlier in Section 3, this shows that non-robust features may not be entirely unintelligible to humans. Our visualizations (Figure 5c) suggests that universal non-robust features may be capturing intuitive properties, such as the presence of a faint bird head. Thus, non-robustness in networks may partially arise from relying on cues that are small in magnitude but still human-aligned.

---

[1] We also provide alternate analyses in Appendix A.3 to bound robust feature leakage in alternate ways, based on extending the methods in Section 5.

[2] Jetley et al. (2018) use a similar analysis to study subspaces that are essential for classification and also susceptible to adversarial perturbations.

[3] Universal perturbations on a robust model require a larger norm as the model is robust near the $\epsilon$ it is trained at.

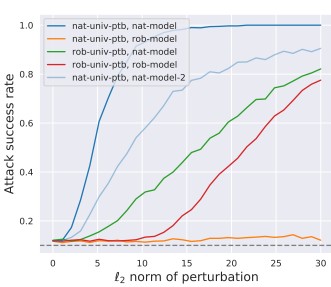
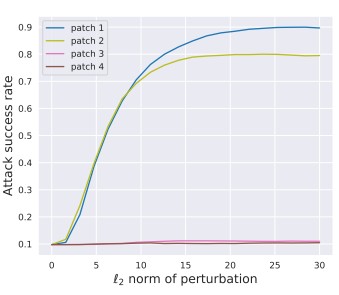
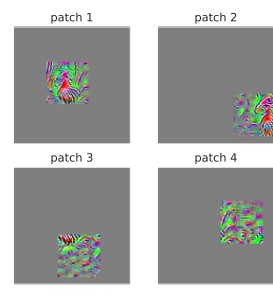

(a) Scaling universal perturbations  (b) Scaling cropped patches  (c) Different random cropped patches

Figure 5: Scaling analysis of universal perturbations on ImageNet-M10: (a) we generate a universal perturbation (`nat-univ-ptb`) of $\ell_2$ norm $\epsilon = 6.0$ on a natural model (`nat-model`) for the target *bird*, and measure the sensitivity of the natural and a robust model (`rob-model`) at various rescalings of the perturbation. We also generate a universal perturbation (`rob-univ-ptb`) of norm $\epsilon = 30.0$ for the robust model and use it to show that the robust model *does* react to scalings of some perturbation. The same `nat-univ-ptb` is also evaluated on an independent natural model (`nat-model-2`) for control; (b) different cropped local patches of the same `nat-univ-ptb` are also evaluated on the natural model; (c) visualizations of the cropped patches.

## 5 UNIVERSAL NON-ROBUST FEATURES CONVEY LESS SIGNAL

So far, we demonstrated that universal perturbations tend to leverage highly semantic non-robust features. We now show that these universal non-robust features have less generalizable signal than standard non-robust features. As in Ilyas et al. (2019), we measure the usefulness of non-robust features based on: (1) how well a model generalizes to the original test set when trained on a dataset where the *only* useful features are those introduced by universal perturbations; and (2) the transferability of universal perturbations across models. (1) directly measures the usefulness of universal non-robust features for learning, while (2) measures the degree to which different models learn the same set of features.

### 5.1 GENERALIZATION FROM UNIVERSAL NON-ROBUST FEATURES

We construct two datasets, $\widehat{\mathcal{D}}_{univ}$ and $\widehat{\mathcal{D}}_{adv}$, where the only useful features are universal non-robust features and non-robust features, respectively.

**Constructing $\widehat{\mathcal{D}}_{adv}$ and $\widehat{\mathcal{D}}_{univ}$.** We follow the procedure outlined in Ilyas et al. (2019) to construct $\widehat{\mathcal{D}}_{adv}$ (which corresponds to $\widehat{\mathcal{D}}_{rand}$ from their paper): 1) for each input-label pair $(x, y)$, select a target class $t$ uniformly at random; then 2) compute an $\ell_2$ adversarial perturbation $\delta$ on $x$ targeted towards class $t$, and replace the original input-label pair $(x, y)$ with $(x + \delta, t)$. By construction, $x$ and $t$ are decorrelated and the only features correlated with $t$ are those leveraged by $\delta$. Because $\delta$ is constrained to be within an $\ell_2$ ball, it can only yield non-robust features.[4] Next, we adapt the above procedure to construct $\widehat{\mathcal{D}}_{univ}$. To this end, in step 2, we use a universal perturbation in place of an adversarial perturbation.[5]

**Comparing the signal in $\widehat{\mathcal{D}}_{adv}$ and $\widehat{\mathcal{D}}_{univ}$.** We train new ResNet-18 models on the $\widehat{\mathcal{D}}_{adv}$ and $\widehat{\mathcal{D}}_{univ}$ datasets and evaluate them on the original test set. [6] The best generalization accuracies from training on $\widehat{\mathcal{D}}_{univ}$ and $\widehat{\mathcal{D}}_{adv}$ were 23.2% and 74.5%, respectively. This indicates that universal non-robust features have much less generalizable signal than standard non-robust features.

---

[4] We address possible robust leakage in Appendix A.3.

[5] There are a few complications in this new setup, one being that universal perturbations only fool a fraction of images. See Appendix A.1 for more details on the construction of $\widehat{\mathcal{D}}_{univ}$.

[6] We report the best results found over a grid of training hyperparameters.

Table 1: Interpolating universality: generalization from $\widehat{\mathcal{D}}_{univ}$ on smaller base sets. For ImageNet-M10, $\ell_2$.

| Size of base set ($K$) | Test Accuracy (%) |
|---|---|
| 1 | 74.5 |
| 2 | 57.1 |
| 4 | 61.3 |
| 8 | 57.4 |
| 16 | 34.3 |
| 32 | 21.8 |
| 256 | 19.1 |

Table 2: Generalization from class-universal and sub-class universal non-robust features on ImageNet-M10, using $\ell_2$ perturbations. We find minimal evidence to suggest that non-robust features are aligned with (sub)classes.

| Source Class | Test Accuracy (%) |
|---|---|
| Random | 23.2 |
| Single Class | 23.9 |
| Single Sub-class | 27.1 |

## 5.2 TRANSFERABILITY OF UNIVERSAL PERTURBATIONS

Ilyas et al. (2019) suggest that the pervasive transferability of adversarial perturbations (Papernot et al., 2016; Moosavi-Dezfooli et al., 2017a) can be attributed to different models relying on similar non-robust features. Consequently, perturbations that leverage more shared non-robust features should transfer better. Here we are primarily interested in transferability as a proxy for detecting shared non-robust features, and we compare the transferability of adversarial and universal perturbations.

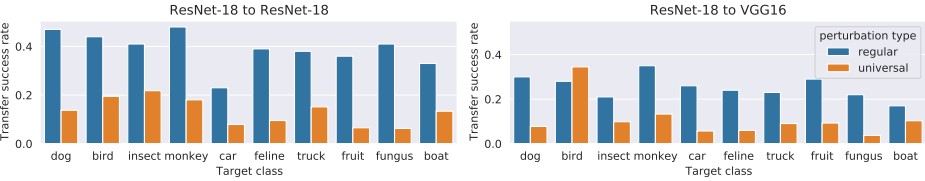

Figure 6: Comparing transferability of adversarial and universal $\ell_2$ perturbations ($\epsilon = 6.0$): perturbations are generated on the source model (ResNet-18), then transferred to a different target model: (left) another ResNet-18 model trained from a different random initialization, and (right) a VGG16 model.

To measure transferability, we (1) perturb examples using either an adversarial perturbation or a universal perturbation on the *source* model, and (2) measure the probability that the perturbed input is classified as the target class on a new *target* model (a ResNet-18 trained independently). To make standard and universal perturbations comparable, we only consider perturbed images that are misclassified by the source model, and also evaluate transfer on images that have labels different from the target label. Figure 6 shows that universal perturbations transfer much less, suggesting that models share fewer universal non-robust features. This is consistent with results from Section 5.1.

## 5.3 INTERPOLATING UNIVERSALITY

The above experiments demonstrate that while universal non-robust features are more human-aligned, they also have less generalizable signal. To explore the underlying trade-off, we investigate to what extent one can interpolate between the properties of universal and standard non-robust features. To this end, we explore two different methods of "interpolating" universality:

**Interpolating via changing the base set size.** One natural way to interpolate universality is to vary $K$, the size of the base set (cf. Section 2.1). $K = 1$ corresponds to adversarial perturbations, and large enough $K$ corresponds to fully universal perturbations. Table 1 shows that generalization behavior generally decreases with the size of the base set. [7] Notably, generalization begins to suffer even for relatively small values of $K$. In contrast, the semantics of universal perturbations improves as the base set becomes larger, in line with the hypothesized trade-off (cf. Appendix A.5).

---

[7] See Appendix A.4 for additional analysis.

**Interpolating via restricting the source class.** We expect that examples from the same class share similar features, including non-robust ones. This motivates us to interpolate universality along a different axis: the level of semantic hierarchy. We consider *class-universal* and *subclass-universal* perturbations, where we restrict the base set to examples from a single class (e.g., dog) or subclass (e.g., Terrier). If certain non-robust features exist only in examples from the same class, we expect class-universal perturbations to leverage those features. To examine this, we repeat the experiments from Section 5.1 using these perturbations (cf. Table 2); the generalization on the original test set improves modestly. This suggests that if non-robust features are specific to small subpopulations of examples, then these subpopulations must be more fine-grained than division into different (sub) classes.

**Discussion** Above results show that we can interpolate the signal present in universal non-robust features by varying the base set's size, and to a smaller extent, its class diversity. This can be intuitively captured by the following toy model: Each example comes with a set of non-robust features, and different examples share a varying subset of features depending on their semantic distance. [8] By computing perturbations over a set of images, we limit the set of non-robust features to their intersection; it follows that the number of features in this intersection, and hence signal, decreases with more images and as the images become further apart.

Existing theoretical models (Tsipras et al., 2019; Allen-Zhu & Li, 2020) fail to distinguish between standard and universal perturbations, and hence fail to capture the phenomena observed here. We believe that more empirical observations are necessary to identify a precise theoretical model.

In Appendix A, we provide additional analyses for some results in this section.

## 6 RELATED WORK

Moosavi-Dezfooli et al. (2017a) first introduce universal adversarial perturbations. Many follow-up works explore its various aspects, including possible theoretical origins, different methods of generation, and defenses (Moosavi-Dezfooli et al., 2017b; Poursaeed et al., 2018; Khrulkov & Oseledets, 2018; Wu & Fu, 2019; Gupta et al., 2019; Hayes & Danezis, 2019; Khrulkov & Oseledets, 2018; Liu et al., 2019; Shafahi et al., 2020).

Using ideas from analysis of universal perturbations, Jetley et al. (2018) find that class-specific patterns unintelligible to humans can induce misclassification while also being essential for classification. Ilyas et al. (2019) first formalize non-robust features, and demonstrate that they alone are sufficient for generalization. In particular, adversarial examples are viewed as a consequence of misalignment between non-robust features and human priors. Conversely, several works discuss the usefulness and perceptual alignment of robust features (Tsipras et al., 2019; Santurkar et al., 2019).

Recent works show how models use certain signals humans do not (primarily) rely on: examples include texture, high-frequency content, and background (Geirhos et al., 2018; Yin et al., 2019; Xiao et al., 2020).

## 7 CONCLUSION

In this work, we revisit universal perturbations and use them to analyze non-robust features. We find that universal perturbations are highly semantic and spatially invariant (in contrast to adversarial perturbations), and give first evidence of perceptually aligned, interpretable non-robust features. However, we also show that they account for only a small part of all non-robust features. Understanding the structure and properties of the remaining bulk of non-robust features is a natural direction for future work. More broadly, developing a framework for studying features in more fine-grained ways—beyond their correlation and robustness—would be helpful for understanding our models better.

---

[8]While the exact definition of distance is hard to define, we can imagine, for example, two images of dogs in the same pose with similar background share more non-robust features than with an image of a car.

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

APPENDICES

In Appendix A, we provide deferred details and analyses for results in Section 5.

In Appendix B, we describe details of the experimental setup and all hyperparameters used in the experiments.

In Appendices C to E, we give a selection of additional results for experiments in the main paper across different settings.

In Appendix F, we discuss the diversity of universal perturbations.

## A  DEFERRED DETAILS FROM SECTION 5

### A.1  DETAILS OF $\widehat{\mathcal{D}}_{univ}$ CONSTRUCTION

We provide additional details for constructing the $\widehat{\mathcal{D}}_{univ}$ dataset in Section 5.1. As it is rather expensive to compute a separate universal perturbation for each input, we compute and use the same universal perturbation per batch of inputs. One possible concern is that re-using same perturbations across the batch results in a reduction in "diversity" of features considered. However, the total number of unique universal perturbations used does not appear to have a large impact from our experience. In particular, using a different universal perturbation for every input does not noticeably affect the generalization of the resulting datasets.

Another complication is that for the robustness threshold $\epsilon$ considered here, universal perturbations fool models only on a fraction $\rho$ of all training examples (while adversarial perturbations at the same threshold succeed on all examples). Hence, we only include examples on which the universal perturbation succeeds (i.e., a pre-trained model classifies the perturbed input as $t$). This filtering step reduces the size of the dataset, so we decrease the number of examples in $\widehat{\mathcal{D}}_{adv}$ accordingly in our evaluation. We also ensure that the distribution of new and original labels is uniform and independent so that the original label is uncorrelated with the new label.

### A.2  ALL RESULTS ON GENERALIZATION ON NON-ROBUST DATASETS

Here we collect additional results for the generalization experiment across various settings: $\ell_2, \ell_\infty$ perturbations in ImageNet-M10 and $\ell_2$ perturbations on on CIFAR-10. We did not conduct experiments on ImageNet, as the constructions of the datasets is computationally too intensive.

Table 3 shows that there is consistently much lower generalization from $\widehat{\mathcal{D}}_{univ}$ than from $\widehat{\mathcal{D}}_{adv}$.

Table 3: Generalization from constructed datasets.

| Source Dataset | Perturbation Set | Constructed Dataset | Test Accuracy (%) |
|---|---|---|---|
| ImageNet-M10 | $\ell_2, \varepsilon = 6.0$ | $\widehat{\mathcal{D}}_{adv}$ | 74.5 |
| ImageNet-M10 | $\ell_2, \varepsilon = 3.0$ | $\widehat{\mathcal{D}}_{adv}$ | 76.6 |
| ImageNet-M10 | $\ell_2, \varepsilon = 6.0$ | $\widehat{\mathcal{D}}_{univ}$ | 23.2 |
| ImageNet-M10 | $\ell_\infty, \varepsilon = 8/255$ | $\widehat{\mathcal{D}}_{adv}$ | 78.7 |
| ImageNet-M10 | $\ell_\infty, \varepsilon = 8/255$ | $\widehat{\mathcal{D}}_{univ}$ | 26.5 |
| CIFAR-10 | $\ell_2, \varepsilon = 1.0$ | $\widehat{\mathcal{D}}_{adv}$ | 64.3 |
| CIFAR-10 | $\ell_2, \varepsilon = 1.0$ | $\widehat{\mathcal{D}}_{univ}$ | 23.3 |

### A.3  BOUNDING ROBUST FEATURE LEAKAGE

The accuracies on the $\widehat{\mathcal{D}}_{univ}$ datasets are very low (for comparison, they are even lower compared to ResNet-18 model trained on (fixed) randomly initialized features, which achieve an accuracy greater than 30%). Given this, one concern is that the small signal we observe can be entirely accounted for by leakage of robust features into these datasets (Ilyas et al., 2019; Goh, 2019).

We run several additional experiments to bound the amount of leakage (we focus on $\ell_\infty$ perturbations on ImageNet-M10):

- We construct a dataset similar to $\widehat{\mathcal{D}}_{det}$ (Ilyas et al., 2019) but instead using universal perturbations, where we choose new corrupted labels by cyclically shifting original labels, rather than choosing them randomly; the robust features now point *away* from the label. Models trained on this new dataset achieve an accuracy up to 19.1% on the original test set, which is less than 26.5% from $\widehat{\mathcal{D}}_{univ}$, but still well above chance-level of 10%. This demonstrates that there is residual signal in universal perturbations that cannot be entirely accounted for by robust leakage.

- We can also probe the original dataset by seeing how well different features[9] can fit the dataset. Adapting the procedure in Goh (2019), we take pre-trained features from different natural and $\ell_\infty$ robust models on ImageNet-M10, and train a linear classifier over those features on $\widehat{\mathcal{D}}_{univ}$. The results in Table 4 show that features from natural models, even sourced from a different model than the one used to generate $\widehat{\mathcal{D}}_{univ}$ or from a different architecture, capture more of the signal in $\widehat{\mathcal{D}}_{univ}$ than robust features. For comparison, the model trained on $\widehat{\mathcal{D}}_{univ}$ from scratch, as we saw earlier, achieves an accuracy of 26.5%, which is higher than 22.3%. This demonstrates again that while some leakage is happening, it cannot account for all of the signal in universal perturbations.

Table 4: Training on $\widehat{\mathcal{D}}_{univ}$ using fixed features to check for leakage.

| Feature Source | (Fine-tuned) Test Accuracy (%) |
|---|---|
| natural | 36.9 |
| natural, diff init | 35.0 |
| natural, diff arch (VGG16) | 26.4 |
| robust ($\ell_\infty$) | 22.3 |

- The scaling analysis from Section 4 also gives supporting evidence, from a different angle.

These results consistently show that while some robust leakage occurs, it cannot account for all of the signal in universal perturbations. A small but non-trivial amount of signal comes from the universal non-robust features.

### A.4    ADDITIONAL ANALYSIS OF $K$-INTERPOLATION

One plausible hypothesis for the degradation of signal quality between $K = 1$ and higher $K$ is that optimizing the perturbation over more images splits the effective perturbation norm budget, and is akin to optimizing over each image separately with a reduced budget, say $\epsilon/K$. To test this hypothesis, we constructed $\widehat{\mathcal{D}}_{adv}$ with a reduced budget of $\epsilon = 3.0$. We compare the generalization from this new dataset to that from the $K = 2, \epsilon = 6.0$ case in Table 5 (latter is the same value as in Table 1). Given the large gap, it seems unlikely that this simple norm budget sharing can explain the above interpolation phenomena. This suggests that when universal perturbations are computed over multiple images, different non-robust features in different images are interacting in a non-trivial manner.

Table 5: Comparing generalization from two different constructed datasets for ImageNet-M10 using $\ell_2$ perturbations. $K$ is the base set size; $K = 1$ corresponds to using standard adversarial perturbations, and $K = 2$ to computing perturbations over random pairs of images.

| Perturbation Type | Test Accuracy on Original (%) |
|---|---|
| $\epsilon = 6.0, K = 2$ | 57.1 |
| $\epsilon = 3.0, K = 1$ | 76.6 |

---

[9]As usual, we consider the penultimate layer representations of the network before the final linear classifier.

## A.5 VISUALS OF INTERPOLATING UNIVERSALITY



Figure 7: Visualization of universal perturbations for target *bird*, computed over base sets of different size $K$. We observe that the visual quality improves with the size of the base set.

Table 1 shows that generalization decreases with the size of the base set. In contrast, Figure 7 shows that the semantic quality of universal perturbations improves as the base set becomes larger. This suggests a possible trade-off between semantic quality and generalizable signal, at least along this particular axis. However, a more careful examination suggests that the picture might be more nuanced: better semantics only become obvious at higher values of $K$ ($K \gtrsim 64$), whereas generalization suffers even at relatively small values of $K$ ($K \sim 16$). Understanding whether these trends are both manifestations of the same underlying phenomena require further work.

## A.6 ALTERNATE BASE SETS

In this paper, we focused on computing universal perturbations over base sets (cf. Section 2.1) consisting of different images (from possibly a restricted set of classes). To study the importance of sample diversity in the base set, we computed universal perturbations over the base set consisting of $K$ different augmentations of a *single* image. The resulting universal perturbations still have similar (but less salient) visual characteristics, and are transferable but to a lesser degree than perturbations computed over base set of different images.

# B EXPERIMENTAL SETUP

## B.1 DATASETS

Our experiments use CIFAR-10 and ImageNet-M10, a subset of ImageNet ILSVRC2012 (Russakovsky et al., 2015); we focus primarily on ImageNet-M10.

ImageNet-M10 consists of ten super-classes, each corresponding a WordNet (Miller, 1995) ID in the hierarchy. Different super-classes contain varying number of ImageNet classes, so the dataset is balanced by choosing six classes within each super-class. The class numbers of the corresponding ImageNet classes are shown in Table 6 For most experiments, the super-classes correspond to the ten labels; for experiments using fine-grained classes, we use the 60 (sub) classes as labels. A sample image from each class is shown in Figure 8

The ImageNet-M10 dataset can be used easily on top of a standard ImageNet directory by loading the `mixed_10` dataset from the `robustness` library (Engstrom et al., 2019).

ImageNet-M10 is also similar to restricted ImageNet, which has been used for adversarial robustness research (Ilyas et al., 2019; Santurkar et al., 2019).

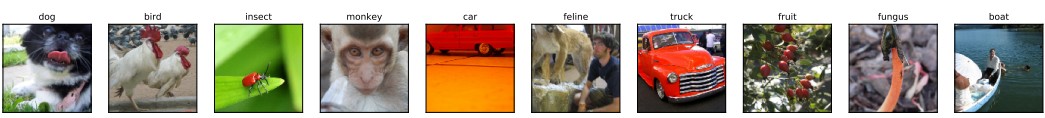

Figure 8: Sample images from each (super) class of ImageNet-M10.

Table 6: ImageNet classes used in the ImageNet-M10 dataset.

| Class | WordNet ID | Corresponding ImageNet Classes |
|-------|-----------|-------------------------------|
| "Dog" | n02084071 | 151 to 156 |
| "Bird" | n01503061 | 7 to 12 |
| "Insect" | n02159955 | 300 to 305 |
| "Monkey" | n02484322 | 370 to 375 |
| "Car" | n02958343 | 407, 436, 468, 511, 609, 627 |
| "Feline" | n02120997 | 286 to 291 |
| "Truck" | n04490091 | 555, 569, 675, 717, 734, 864 |
| "Fruit" | n13134947 | 948, 984, 987 to 990 |
| "Fungus" | n12992868 | 991 to 996 |
| "Boat" | n02858304 | 472, 554, 576, 625, 814, 914 |

## B.2 HYPERPARAMETERS

All experiments in the paper involve a combination of: training a model, computing adversarial perturbations, and computing universal perturbations. Below, we provide details for all the hyperparameters and their selection.

**Robustness threshold** We use the same robustness threshold throughout: $\ell_2, \epsilon = 6.0$ and $\ell_\infty, \epsilon = 8/255$ for ImageNet datasets, and $\ell_2, \epsilon = 1.0$ for CIFAR-10.

### B.2.1 TRAINING MODELS

**Training base models** All models are trained with SGD with momentum and standard data augmentation. For corresponding robust models, the same optimization parameters are used with PGD training (Madry et al., 2017) using 3 steps and attack step size $\frac{2}{3}\epsilon$ (for both $\ell_2$ and $\ell_\infty$). Weight decay of 5e-4 was used in all cases. The accuracies of these models are shown in Table 7.

Table 7: Standard and robust models used in our experiments.

| Dataset | Model | Standard Test Accuracy (%) | Robust Test Accuracy (%) |
|---------|-------|---------------------------|--------------------------|
| ImageNet-M10 | Standard | 95.7 | $< 1.0$ |
| | $\ell_2$ | 86.7 | 59.8 |
| | $\ell_\infty$ | 87.6 | 59.2 |
| CIFAR-10 | Standard | 94.8 | $< 1.0$ |
| | $\ell_2$ | 80.0 | 50.7 |

**Training models on constructed datasets** The models on the constructed datasets, including $\widehat{\mathcal{D}}_{univ}$ and $\widehat{\mathcal{D}}_{adv}$ (Section 5.1), are trained over the following grid of hyperparameters: three learning rates (0.01, 0.05, 0.1), two batch sizes (128, 256), including/ not including a single learning rate drop by a factor of 10. All models are trained with 400 epochs, standard data augmentation, and weight decay $5 \cdot 10^{-4}$.[10] We report the setting corresponding to the highest test accuracies in Table 8.

As an exception, for the $K$-interpolated datasets (Section 5.3), we fixed a single set of hyperparameters to train models.

### B.2.2 COMPUTING ADVERSARIAL PERTURBATIONS

**Adversarial perturbations** For a given $\epsilon$, PGD with an attack step size of $\frac{\epsilon}{3}$ and 10 steps is used (no random restarts.)

---

[10]Without data augmentation, the model overfit on these constructed datasets.

Table 8: Default and best hyperparameters for training models on different datasets.

| Source Dataset | Constructed Dataset | Epochs | LR | Batch Size | LR Drop |
|---|---|---|---|---|---|
| ImageNet-M10 | $\mathcal{D}$ (original) | 200 | 0.1 | 256 | 50, 100, 150 |
| | $\widehat{\mathcal{D}}_{adv}$ ($\ell_2$) | 400 | 0.01 | 128 | 250 |
| | $\widehat{\mathcal{D}}_{univ}$ ($\ell_2$, random class) | 400 | 0.01 | 128 | 250 |
| | $\widehat{\mathcal{D}}_{univ}$ ($\ell_2$, $K$-interp) | 400 | 0.01 | 256 | 250 |
| | $\widehat{\mathcal{D}}_{univ}$ ($\ell_2$, same class) | 400 | 0.05 | 64 | None |
| | $\widehat{\mathcal{D}}_{univ}$ ($\ell_2$, same subclass) | 400 | 0.05 | 128 | 250 |
| | $\widehat{\mathcal{D}}_{adv}$ ($\ell_\infty$) | 400 | 0.001 | 256 | None |
| | $\widehat{\mathcal{D}}_{univ}$ ($\ell_\infty$) | 400 | 0.1 | 128 | None |
| CIFAR-10 | $\mathcal{D}$ (original) | 150 | 0.1 | 128 | 50, 100 |
| | $\widehat{\mathcal{D}}_{adv}$ ($\ell_2$) | 400 | 0.05 | 256 | 250 |
| | $\widehat{\mathcal{D}}_{univ}$ ($\ell_2$) | 400 | 0.05 | 256 | None |

**Universal perturbations** The hyperparameters of the PGD algorithm for universal perturbations are: learning rate schedule[11], number of epochs, and batch size.[12] In order to see the impact of the choice of these parameters, we ran a grid search over the parameters; a sample of these selections is visible in Figure 9. Overall, we observe that the exact choice of these parameters did not have a large impact on the final training accuracy (and the quality of the resulting perturbations, both in terms of visuals and generalization to other images) as long as (1) the initial learning rate is not too low or too high, and (2) a sufficient number of epochs is used. So in all of our experiments, we fix a learning rate of 2.0 and a batch size of 128 or 256, and trained a sufficient number of epochs (max of 100).[13] The use of a learning rate decay did not seem to have much impact, so we use the same learning rate throughout.

## C  VISUALIZATIONS

We show visualizations of universal perturbations across different datasets (CIFAR-10, ImageNet-M, and ImageNet), architectures (VGG for Mixed10), and norm constraint ($\ell_\infty$ for Mixed10).

**Normalization for visualization.**   For all visualizations, perturbations were rescaled as follows (for each input channel): first truncated to $[-3\sigma, 3\sigma]$ (where $\sigma$ is the standard deviation across all channels and pixels), then rescaled and shifted to lie in $[0, 1]$.

### C.1  DIFFERENT DATASETS

---

[11]Sometimes this is known as step size in the adversarial examples literature.

[12]Batch size only matters if it is smaller than the size of the base set, $K$.

[13]In this case $K$ is much larger than batch size, fewer epochs suffice as the effective number of updates increases due to bounded batch size. In particular, for a fully universal base set (e.g. the entire training set), as few as five epochs seed to suffice.

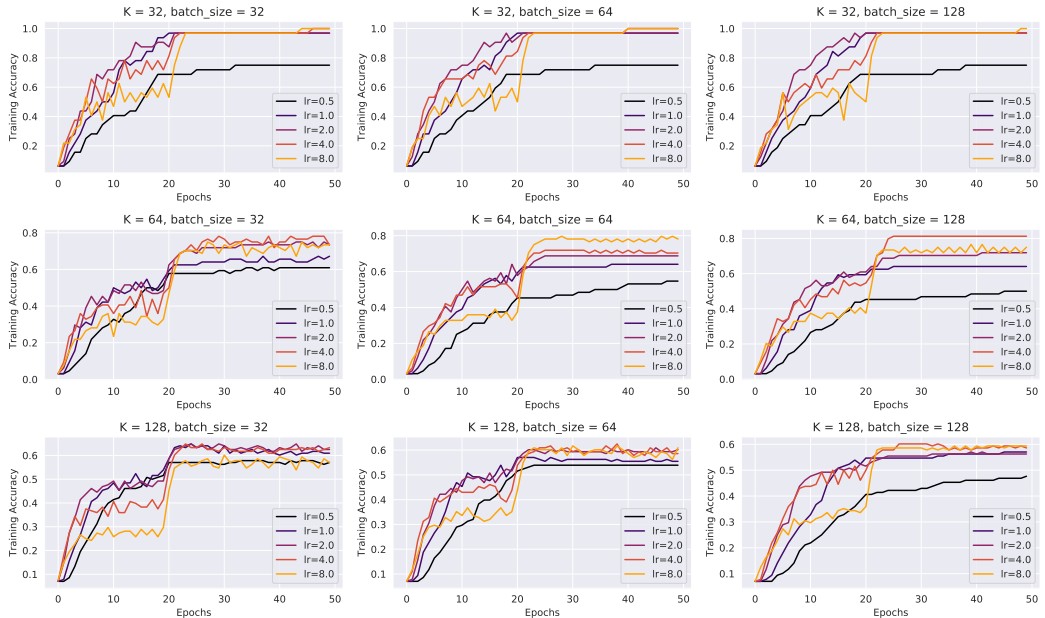

Figure 9: The learning dynamics of computing universal perturbation (epoch vs. training accuracy on the base set) are shown across a selection of learning rates, batch sizes, and three intermediate values of $K$.

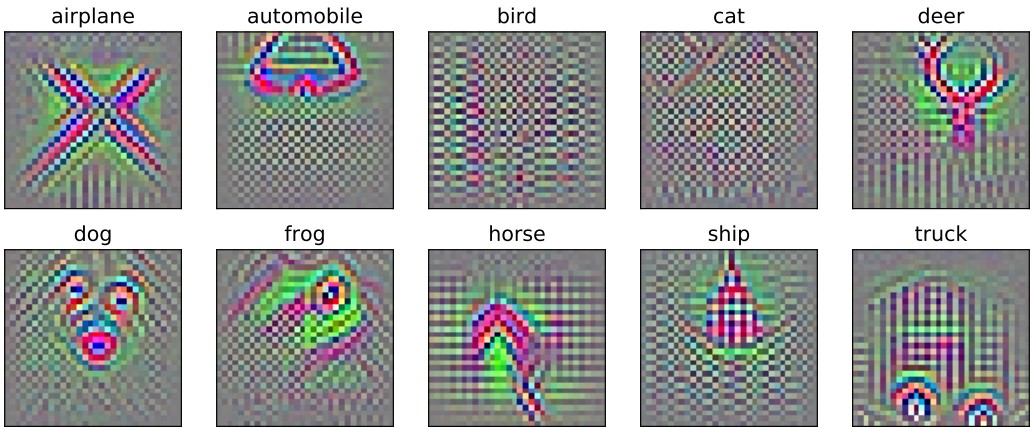

Figure 10: Visualization of $\ell_2$ universal perturbations at $\epsilon = 1.0$ for a **CIFAR-10** model. Optimization on some classes (bird and cat) fail occasionally, and is reflected in their lack of semantic content.

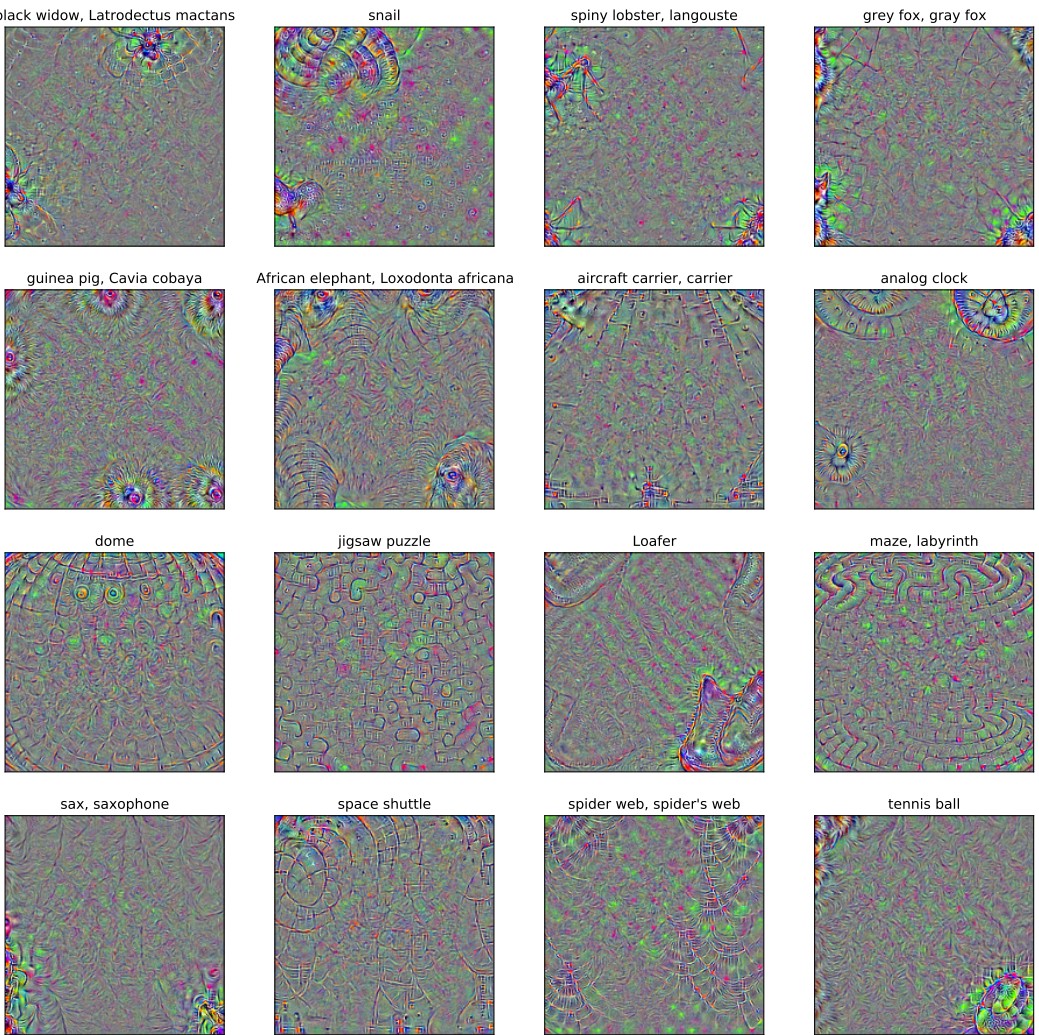

Figure 11: Visualization of $\ell_2$ universal perturbations at $\epsilon = 6.0$ for a full **ImageNet** model. We hand-picked the classes with the most salient perturbations. Many of the perturbations possess visual characteristics identifiable with the corresponding class.

## C.2 OTHER ARCHITECTURES

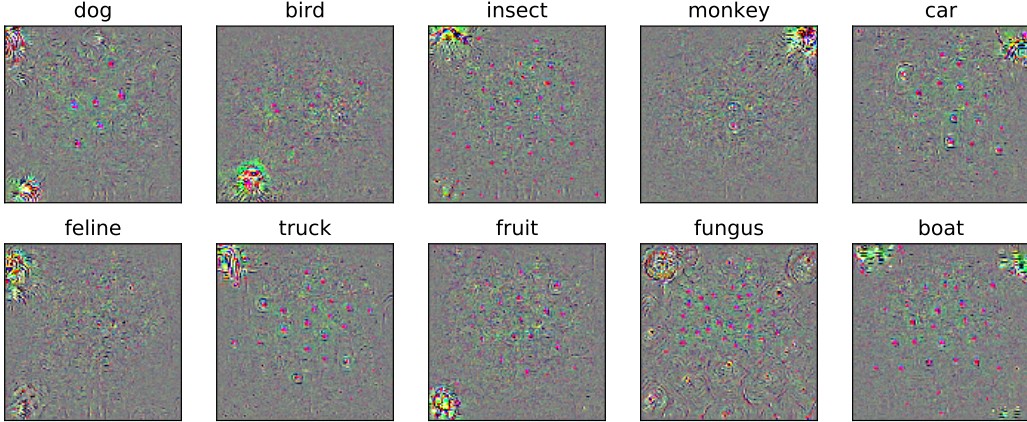

Figure 12: Visualization of $\ell_2$ universal ($K = 512$) perturbations at $\epsilon = 6.0$ for ImageNet-M10, generated this time on a **VGG11** network (Simonyan & Zisserman, 2015).

## C.3 $\ell_\infty$ PERTURBATIONS

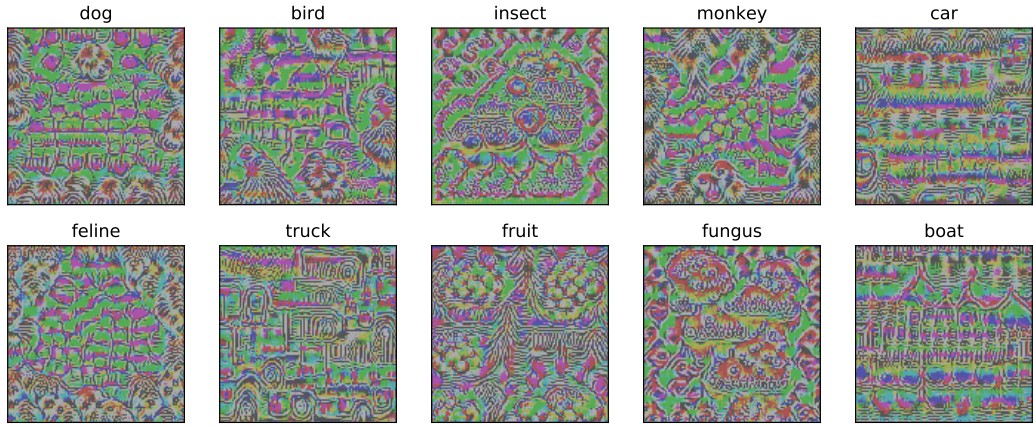

Figure 13: Visualization of $\ell_\infty$ universal perturbations at $\epsilon = 8/255$ for different target classes. We observe recognizable patterns for each class similar to those of $\ell_2$ universal perturbations, but the remaining regions have much higher energy and different texture.

## D  LOCALITY ANALYSIS

We provide full results of randomized locality analysis on ImageNet-M10 in Figures 14 and 15. We did not perform the analysis on CIFAR-10 perturbations, as the perturbations are more "global" by nature, e.g. the semantic part seems to already occupy most of the perturbation.

**Exploring properties of $\ell_2$ vs $\ell_\infty$ perturbations.** For both $\ell_2$ and $\ell_\infty$ universal perturbations, the patches that are most identifiable with their classes have much higher ASR. A major difference is that for $\ell_2$ perturbations, the patches with low ASR have much smaller norms, whereas for $\ell_\infty$ perturbations, the patches with low ASR have the similar norms (measured in either $\ell_2$ or $\ell_\infty$). This is expected, as for $\ell_2$ every local patch contributes to the total $\ell_2$ norm and it is unsurprising that optimization avoid putting weight on patches with little signal; in contrast, for $\ell_\infty$ perturbations, different patches are independent in meeting the norm bound, so the perturbation can have patches

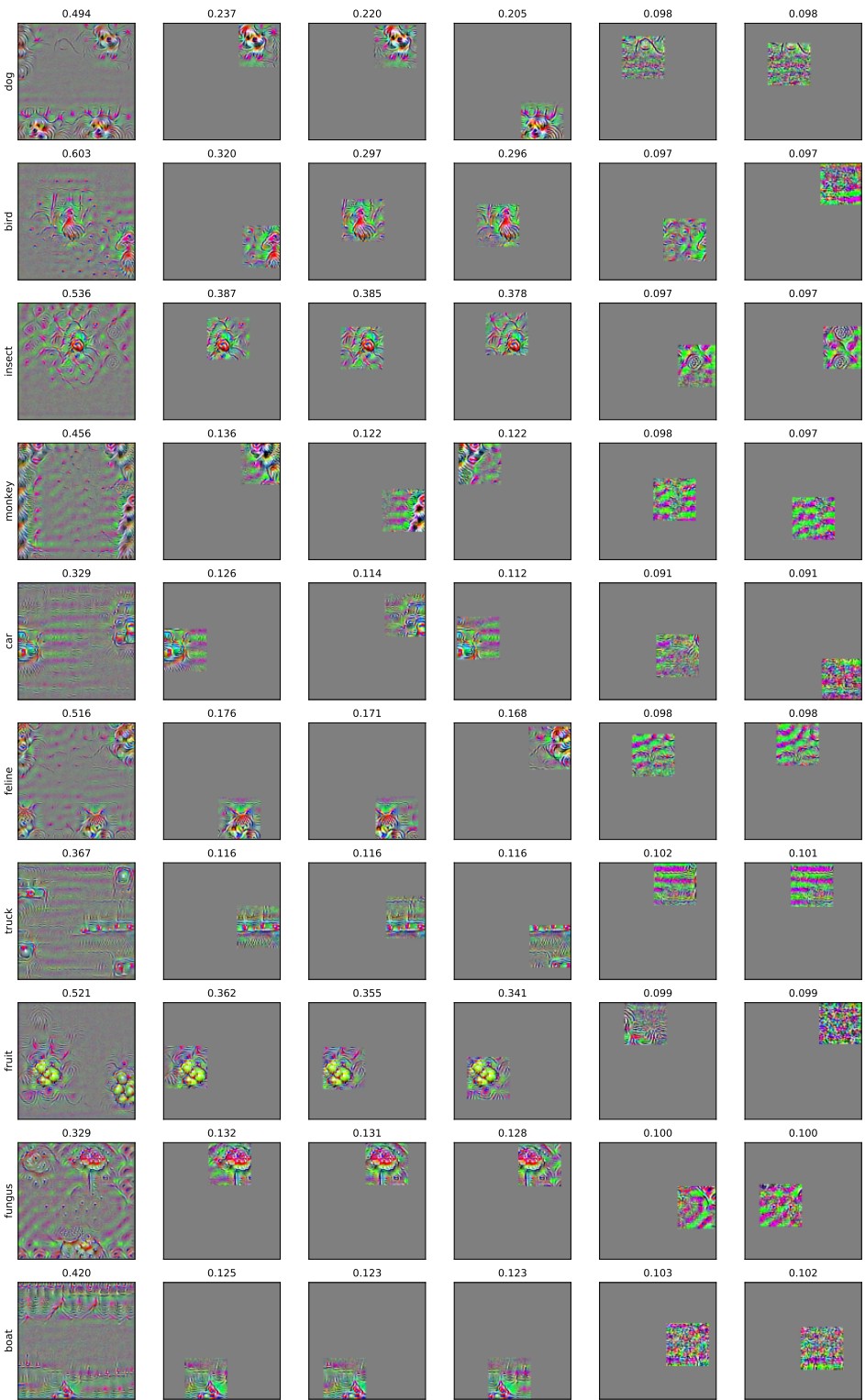

Figure 14: Locality analysis of $\ell_2$ universal perturbations on ImageNet-M10. Each row shows the perturbation for the particular class, followed by three patches with highest ASR and two patches with lowest ASR; ASR is written above each patch.

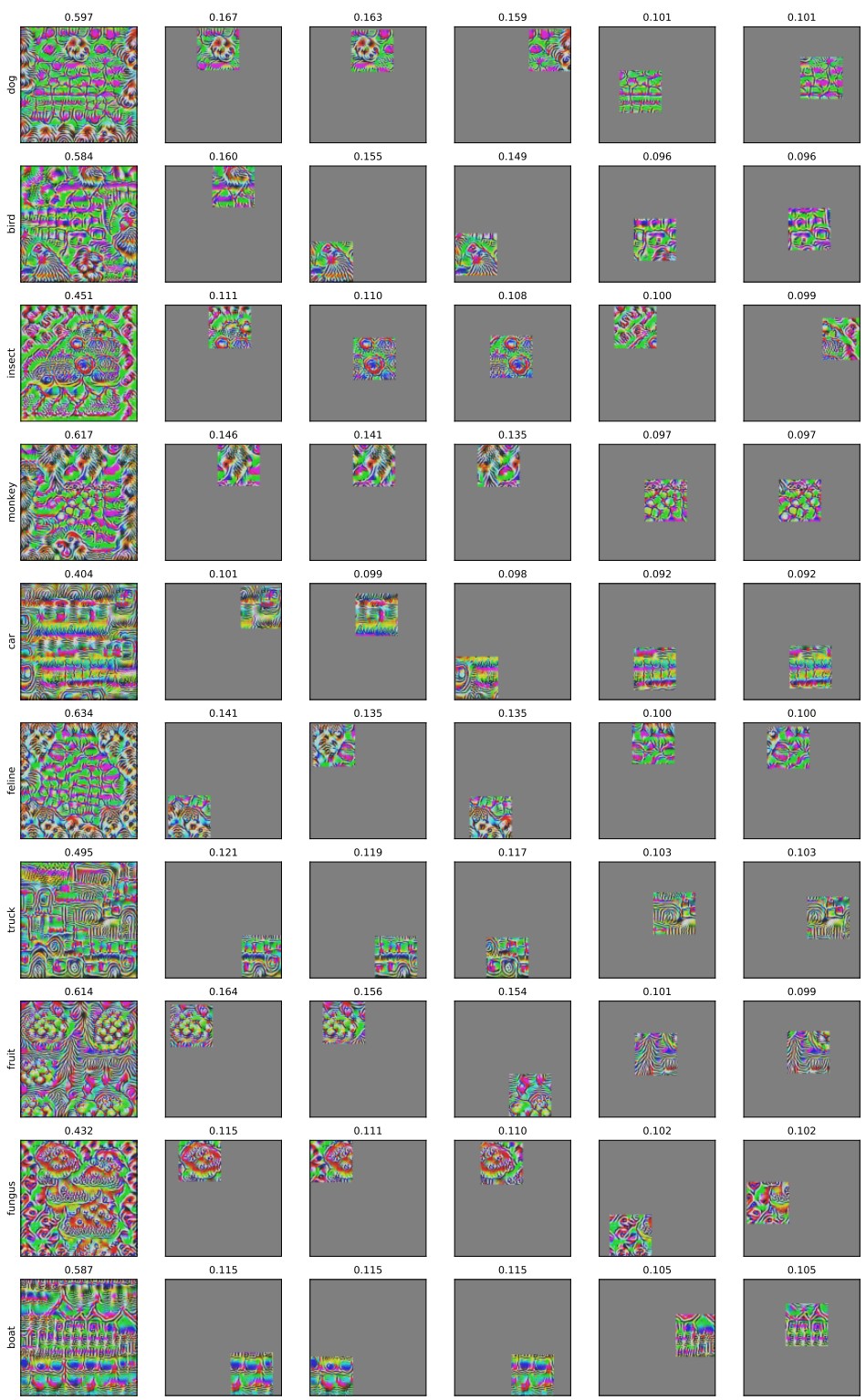

Figure 15: Locality analysis of $\ell_\infty$ universal perturbations on ImageNet-M10. Each row shows the perturbation for the particular class, followed by three patches with highest ASR and two patches with lowest ASR; ASR is written above each patch

with low signal that are similar in norm. However, it is not clear whether these low signal patches found by optimization serve some purpose. To test this, we experimented with zeroing-out these patches or replacing them with copies of patches with the highest ASR. Evaluation of the resulting perturbations show that the ASR is highest with these patches in place, and are in fact lowered when we replace them with patches with higher ASR. This suggests that these low signal patches have a non-linear interaction with rest of the perturbation, and are providing a small boost in signal when combined with other patches. Understanding this phenomena in more depth may shed more light on the differences between properties of $\ell_\infty$ and $\ell_2$ universal perturbations.

# E  SPATIAL INVARIANCE

Section 3.2 shows a sample of translation invariance evaluation for perturbations on ImageNet-M10. We include additional samples from evaluation using $\ell_2$ perturbations on CIFAR-10, to check that the phenomena is general.



Figure 16: We evaluate attack success rates of translated adversarial and universal perturbations for different target classes over the test set on CIFAR-10. There is no subsampling of the grid here as the dimensions are much smaller. We used a subsampled grid of all the possible offsets with strides of four pixels. As before, the value at coordinate $(i, j)$ represents the average attack success rate when the perturbations are shifted by $i$ pixels along the $x$-axis and $j$ pixels along the $y$-axis, with wrap around to preserve information. Evaluation of adversarial perturbation is only shown for one target as it is redundant.

# F  PERTURBATION DIVERSITY

Throughout our investigations, we observed that the computed universal perturbations seemed to lack in diversity. We see in Figure 17 (first row) that generating multiple different universal perturbations for a given target leads to visually similar perturbations. [14] Prior works study generating more diverse universal perturbations (Reddy Mopuri et al., 2018).

To increase diversity, we generated universal perturbations with an additional penalty to encourage orthogonality of perturbed representations at different layers of the network.[15] Even with the added regularization, the resulting perturbations appear to be remain similar, but with changes in spatial arrangements of the semantic patterns (second row).

One effective way we found to increase the diversity of these perturbations is to compute universal perturbations for targeting specific *sub-classes* of Mixed10. To do so, we first train a fine-grained model using Mixed10's subclass labels, and generate universal perturbations for each of the six sub-classes per class. In Figure 18, we show the universal perturbations targeting individual sub-classes of the bird class. We observe distinct patterns characteristic of each species of birds, an increase in diversity from before. However, generation of these perturbations requires a model trained on the fine-grained labels. An interesting question is whether one can recover such diversity using just the model trained on coarse-grained labels without injection of additional label information.

---

[14]In the original work on universal perturbations, Moosavi-Dezfooli et al. (2017a) made the opposite interpretation that the perturbations generated independently are diverse due to their inner product being small. However, it is unclear whether inner product in the input space is the appropriate measure of their similarity. In fact, given the evidence of weak signal in universal perturbations (Section 5), it appears more likely that different perturbations are highly similar in their (signal) content.

[15]For an extensive survey of similar and other regularization techniques for visualization, see Olah et al. (2017).

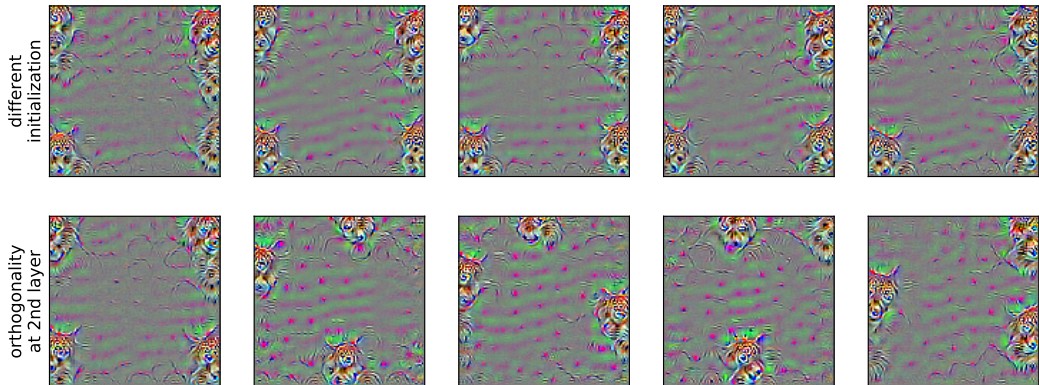

Figure 17: Visualization of $\ell_2$ universal perturbations for class feline, computed using different initializations (first row), and an additional penalty to encourage diversity (second row).

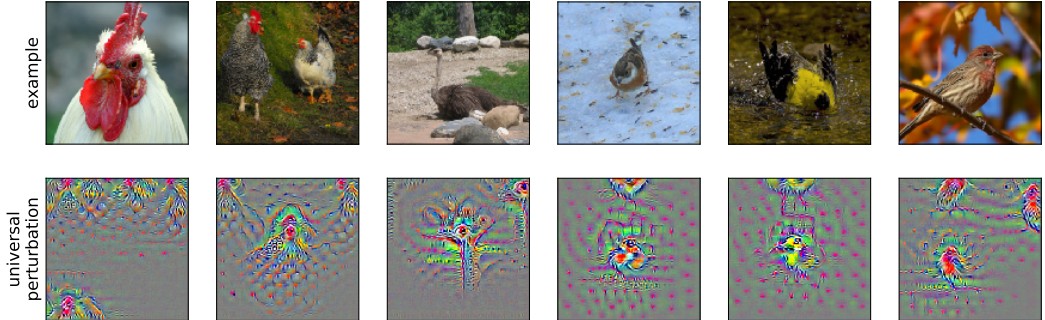

Figure 18: Visualization of $\ell_2$ universal perturbations for six individual sub-classes inside the *bird* class. These were generated on a *fine-grained* model trained on the subclass labels (60 in total). Observe the distinctive visual characteristics of each class, for instance the outline of a chicken (column 2), the neck of an ostrich (column 3), and the distinctive colors of different species (columns 3-5).

