# OpenReview forum: "Non-robust Features through the Lens of Universal Perturbations"
_ICLR.cc/2021/Conference — Reject_

### Official Review · AnonReviewer2 · 2020-10-27
**This paper provides detailed analysis of the property of universarial adversarial attacks. It is interesting to study non-robust features and universal perturbations together.**

**Rating:** 5
**Confidence:** 4

**Review:**

This paper analyzes the existence of non-robust features through the lens of universal perturbations. The concept of non-robust features and universal perturbations is not new, but it is interesting to study the two concepts together.

Strength: the paper gives a detailed study of the property of non-robust features via universal perturbations. Visualization of qualitative results is provided.

Weakness:

1. Conclusion is not well supported by the experiment. It is hard to measure what is human-aligned properties, even the concept is not well-defined. By visualization and spatially invariant experiments, it is fine to hint towards this, but seems unconvincing to me.

2. The paper does not offer related improvements for existing models. How would the findings of this paper be beneficial to the models? Or guide the research community to build more robust models, or launch successful attacks?

3. The findings are not that interesting and kind of known to the community. Wouldn't universal perturbations be optimized to be invariant such that they contain less non-robust signal (but more invariant/general signal), otherwise how would it be universal?

4. The paper defines the adversarial perturbations to be non-robust features, however, the existence of non-robustness features is still an open question. It is one interpretation, but other interpretation also works. For example, it can be the common vulnerability of CNN models, thus the authors need to rethink whether a paper based on this assumption is solid.

---

> ### Author Response · Authors · 2020-11-17
> **Response**
>
> Thank you for the valuable feedback!
>
> We address each of the concerns below:
>
> 1) We agree that defining and measuring human-alignedness is a non-trivial task, as interpretability in general is a hard, and often ill-defined, problem.
>
>  That said, as long as we define a particular metric or dimension, we can still measure accordingly to draw meaningful conclusions. We believe that our experiments are well-defined and provide sufficient evidence to support our claims about human-alignedness, at least along the two dimensions we study:
>
>     - For the first experiment, showing “predictive signals are localized to semantic patches”, we use normalization and randomization so that the selection of patches is unbiased. Judging visual saliency of the resulting patches inevitably requires human judgement. Given the persistent pattern we have observed, we decided there might be no need for even more rigorous crowd-sourced study, but we would welcome the reviewers perspective on this.
>
>     - For the translation invariance experiment, there is no subjectivity involved at all, as we are measuring models’ accuracies across different (shifted) perturbations; the comparisons are entirely quantitative. If we accept translation invariance as a human-aligned property, then our experiment gives evidence for (significant) differences between universal and regular adversarial perturbations.
>
>  We edited our draft to be more consistent and precise in our discussion of semantics and human-alignedness, and we hope that this addresses this useful feedback.
>
> 2) We think that improving models’ adversarial robustness or finding better attacks is beyond the scope of this paper. (As we mentioned above, our goal is to probe the nature of non-robust features rather than explore the ML security aspects of this phenomenon.) That said, our study uncovers new aspects of non-robust features, which had been shown to be critically tied to our model’s susceptibility to adversarial examples. We believe that our findings here provide a starting point for improving our understanding of non-robust features (which have remained fairly abstract until now), and that such an understanding is important for building more robust models that rely less on these features.
>
> 3) While some previous works have made related observations regarding the semantics of universal perturbation, all of our specific findings are new (*). In particular, as far as we can tell, our focus on using universal perturbations to shed light on non-robust features is a completely new approach.
>
>  Re: “Wouldn’t universal perturbations be optimized...to contain less non-robust signal.” We agree with this intuition and discuss this in Section 5.3. While our findings here may not be entirely surprising, we find our analysis and its implications to be valuable to improving our understanding of non-robust features.
>
>   (*) In our updated draft, we have added discussion of previous works throughout the paper to clarify connections and comparisons to prior works.
>
> 4) Even though we believe that the experiments of Ilyas et al. give strong evidence for existence of non-robust features, we agree that their existence does *not* rule out other reasons that can give rise to adversarial vulnerability. Indeed, follow-up works such as Nakkiran 2019 (https://distill.pub/2019/advex-bugs-discussion/response-5/) show that one can find adversarial examples which do not seem to leverage non-robust features. Also, as the reviewer points out, adversarial examples may also arise due to a particular vulnerability of CNN models, or any of the other reasons previously suggested in the literature (high dimensionality, insufficient data, etc.)
>
>  Exactly for these reasons, we make sure to not define adversarial perturbations to be non-robust features, and we explicitly take care to distinguish between them throughout writing (for specific definitions, see Preliminaries). In particular, we make sure to emphasize that there’s a distinction between perturbations (which live in the image space, within the perturbation set), and features leveraged by those perturbations (which are functions on the image space).
>
>  In any case, we do not assume the existence of non-robust features a priori, and use experiments wherever necessary (e.g. Section 5) to provide evidence.

---

### Official Review · AnonReviewer3 · 2020-10-27

**Rating:** 5
**Confidence:** 2

**Review:**

-> Summary:

The authors investigate universal perturbations for adversarial robustness. They find that while universal perturbations are based on non-robust features, they are more human-aligned and spatially invariant. They also show that they contain less predictive
signal than other non-robust features.

-> Reasons for score:

I recommend weak reject for this paper, as I think the scope and impact of this work is too limited to universal perturbations. While I am not that familiar with interpretability literature, what experiments on generalization seem to show, most of the features learned by models are in fact not universal perturbations. So I am not sure whether findings of this paper would be interesting for broad ICLR community.

-> Pros:

I like investigation of the properties of non-robust features in the context of interpretability by humans. Paper is generally well written and easy to follow. Also, findings in this paper are novel as far as I know.

-> Cons:

While I am not that familiar with interpretability literature, based on my knowledge, I am not so sure of the impact of the findings presented in this work. There are essentially three main findings in this paper: universal perturbations are human aligned, non-robust features can be semantic and universal perturbations contain less signal than other non-robust features. I feel these are all interesting, but I am not sure how surprising they are. Universal perturbations are just a small type of possible adversarial perturbations and it is interesting to investigate their properties. As the experiments here show, most of the non-robust features, the ones that are actually useful for classification, are in fact not universal perturbations. So I am not sure what would be the key takeaway for the broader community that does not work on universal perturbations.

---

> ### Author Response · Authors · 2020-11-17
> **Response**
>
> Thank you for the valuable feedback!
>
> The individual findings may not appear surprising, but overall we want to communicate a simple but important new finding: there exists human-aligned non-robust features. While we focus on universal perturbations and their properties, we view this more as a tool to illustrate our main point.
>
> As you noted, the experiments here show that most of the useful non-robust features are still unidentified. We believe that our experiments, and in particular our experiments on interpolating universality, provide some insights on as well as illustrate barriers to identifying and understanding the remaining bulk of non-robust features.

---

### Official Review · AnonReviewer4 · 2020-10-28
**Novel insights into the properties of non-robust features**

**Rating:** 6
**Confidence:** 4

**Review:**

Summary:
This paper studies the link between non-robust features and universal adversarial perturbations. This paper shows that universal perturbation leverage non-robust features in data in a different way than standard adversarial attacks. Experiments are based on a universal version of projected gradient descent (PGD). The findings are that universal perturbations are more aligned with visual semantics and human perception that general adversarial attacks. Moreover, it is shown to be difficult to obtain generalisation or transferability between models based on universal signals, as opposed to standard adversarial samples. Generalization seems to decrease with the size of the set used for generating a given universal perturbation, while semantics of the features improve.

Strong points:
- The paper uncovers novel, previously unknown properties of non-robust features.
- The subject of the paper is of interest in general and relevant for the ICLR community.
- The experiments chosen for the exploration are appropriate.
- The paper is clear and well-written.

Concerns:
- While the present work advanced the state of understanding of non-robust features, it seems we are still far from a full picture of the topic. Many of the novel ideas proposed in the paper are still at a hypothesis or reasonable explanation state (e.g., universal perturbations reflect the intersection of non-robust features for the input set they were computed on).

Questions / suggestions:
- It is unclear why universal perturbations being bounded in norm implies that they must be leveraging non-robust features.
- Fig. 4: how is the perturbation shifted? Is is rolled over the edge of the image or is the excess discarded?
- The current work can probably be extended to a similar analysis on adversarial patches.

---

> ### Author Response · Authors · 2020-11-17
> **Response**
>
> Thank you for the valuable feedback!
>
> We address each of the comments below:
>
> * Re: "While the present work advanced the state of understanding of non-robust features, it seems we are still far from a full picture of the topic..."
>
>  We agree that we are still far from a full picture of non-robust features — attaining a full grasp of these turns out to be a rather difficult (and so far elusive) goal. Still, we believe that our work constitutes a promising step towards tackling this challenge.
>
> * Re: "It is unclear why universal perturbations being bounded in norm implies that they must be leveraging non-robust features."
>
>  Universal perturbation being bounded in norm only suggests that they are likely to leverage non-robust features (as non-robust features are by definition features that can be “flipped” within the perturbation budget). Indeed, as we discussed in Section 4, there is a possibility of robust feature leakage. However, our experiments in that section and experiments in the Appendix A.3 show that this contribution is limited, so we have evidence that these universal perturbations are leveraging non-robust features.
>
>  We have also edited our writing to improve clarity and added additional analyses to Appendix A.3.
>
> * Re: "Fig. 4: how is the perturbation shifted? Is is rolled over the edge of the image or is the excess discarded?"
>
>  The perturbation is wrapped around, e.g. rolled over, in order to preserve information.
>
> * Re: "The current work can probably be extended to a similar analysis on adversarial patches."
>
>  Was there a particular connection the reviewer has in mind relating to adversarial patches?
>
>  Note that adversarial patches are often optimized to be effective across multiple images and under different transformations, e.g. they are designed to be robust to say translation by design (https://arxiv.org/abs/1712.09665). They are unbounded in Lp norm, so they might not be directly comparable to the Lp-bounded universal perturbations considered here.

---

> > ### Comment · AnonReviewer4 · 2020-11-24
> > **Thank you for your response**
> >
> > I would like to thank the authors for the clarifications and the updated draft.
> >
> > To expand on my comment about adversarial patches, this is indeed an idea for an extension. I'm not suggesting a direct comparison between universal perturbations and adversarial patches, for the distinctions that you have mentioned (e.g., different attack budget). Their common property is, as per your response, the "universality" over multiple images of both designs. In that sense, the same analysis tools from the paper could maybe be applied to adversarial patches.

---

> > > ### Author Response · Authors · 2020-11-24
> > > **Thank you**
> > >
> > > Thanks again for the valuable comments and feedback,
> > > and thanks for bringing our attention to the connection to adversarial patches.

---

### Official Review · AnonReviewer1 · 2020-11-05
**an interesting study on universal adversarial perturbations**

**Rating:** 7
**Confidence:** 4

**Review:**

Prior works generally thought non-robust features, which are vulnerable to small perturbations, are not semantically meaningful but are useful for generalization. This work challenges these traditional beliefs by pointing out that non-robust features can also be human-perception aligned and be less useful for generalization, if these non-robust features are discovered via universal adversarial perturbations (rather than via image-dependent perturbations). Extensive experiments are provided to justified these arguments.




*Pros:

(1) This paper is well written and easy to follow.

(2) The idea of utilizing universal adversarial perturbations to analyze non-robust features is novel, and lead to many interesting findings of non-robust features, e.g., non-robust features can also be semantically meaningful.

(3) Most arguments in this paper are rigorously justified by careful experiment designs. For example, by analyzing the effects of perturbation scaling on model performance, this paper successfully persuades the reviewer to believe universal adversarial perturbations indeed leverage non-robust features.



*Cons:

(1) The reviewer has a major concern about the experiments that show non-robust features can also fail to generalize. For example, Table 1 suggests that with a large base set, non-robust features' generalization decrease. Nonetheless, the reviewer worries that this effect may be caused by the reduced diversity of training samples. That's said, when training with a base set of K (i.e., K training samples will share the sample universal perturbation), the reviewer tends to think the training dataset's "effective" sample size will then be greatly reduced to (total sample size)/K, which will decrease training quality and lead to inferior generalization. To rule out this possibility, the authors should also investigate the setting where the number of universal perturbations is equal to the number of training samples, regardless of the value of the size of base set K. The dataset preparation could be like this: first, we randomly sample K images, and generate a universal perturbation for them; second, we randomly sample one image from K' samples (which successfully fool networks), add the generated universal perturbation, and put it to the new training set; third, repeat the first & second step until we collect enough images in this new training set. Only with the performance analysis using these datasets, we can then confidently justify if non-robust features can generalize.

(2) The reviewer is a little bit curious (and worry) why different batch sizes lead to quite different performance as shown in Figure 8? As already shown in the "Training ImageNet in 1hr" paper (https://arxiv.org/pdf/1706.02677.pdf) and many others, if you set the learning rate to be proportional to the batch size, then these training settings should get nearly the same results. Why this paper shows batch size is a critical hyperparameter for impacting model performance (e.g., in Figure 8, the settings of K=64, lr=0.5 batch size=32 and K=64, lr=1.0, batch size=64 seem to get very different results)?

(3) why this paper only considers the targeted attack setting? will the conclusions here generalizable to the setting of non-targeted attack? Also what is the motivation for showing universal perturbations has spatial invariance (since image-dependent perturbation can also be spatial invariance if such transformation is considered by attackers)?

In general, the reviewer thinks it is an interesting paper, and is happy to raise the score if the authors successfully address the concerns above.

---

> ### Author Response · Authors · 2020-11-17
> **Response**
>
> Thank you for the valuable feedback!
>
> We address each of the concerns below:
>
> 1. Reduced diversity may indeed have a possible confounding effect in our raw experimental setup. However, we found that varying the number of distinct perturbations (for fixed K) did not have a noticeable impact on generalization, so it is rather unlikely that effects of feature diversity are significant enough to counter the opposing effects of universality (we mentioned this briefly in Appendix A.1, but did not provide any data). To be clear, these experiments did not test the limiting case where number of perturbations is equal to number of examples.
>
>  To be sure, we ran new experiments, closely following the setup suggested by the reviewer. In particular, we constructed new datasets where each perturbation is applied to exactly once to a unique input. Due to time constraints, we focused on the intermediate datapoints K=4 and K=16 to gauge the trend (note that the construction time increases with K, as we only keep a single example after computing a universal perturbation over K inputs). As we might expect, the generalization increases, but only marginally: for K=4, generalization is 63.5% compared to 61.3%; for K=16, generalization is 38.4%  compared to 34.3% (note that these accuracies are from single training runs so some of the differences can also come from noise). But importantly, the huge gap between K=4 and K=16 persists, showing that generalization indeed decreases with K.
>
>  Based on these new experiments, we can extrapolate and be confident that the gap between D_adv and D_univ is still significant after accounting for confounding effects from reduced diversity of training samples.
>
>  We thank the reviewer for suggesting ways to make our evaluation more rigorous.
>
>
> 2. This is a great observation. However, our setting is quite different from the setting where the learning rate vs batch size scaling behavior is usually studied: here, we are tracking the accuracy (or attack success rate) of a universal perturbation optimized via PGD, whereas for the latter, people look the test accuracy of a model optimized via (minibatch) SGD.
>
>  Both settings can be viewed as instances of general gradient-based optimization. While there is some theoretical motivation for this scaling behavior (https://arxiv.org/abs/1710.06451; https://arxiv.org/abs/1711.04623) that applies generally to minibatch SGD optimization, we suspect the behavior breaks down in our setting for the following reasons:
>     - The scaling behavior derivation uses the approximation that N >> B, where N is the total number of examples and B is the batch size; this usually holds when training models on large datasets with typical batch sizes. In contrast, the settings in Figure 8 do not satisfy this; B is in fact roughly on the same order as N (ex. N=128, B=32).
>     - PGD optimization involves projection steps onto the lp ball, and it is not obvious how the analyses for SGD can incorporate this. In particular, the step size of PGD, or the learning rate, is relatively large compared to the lp ball radius, so “boundary effects” may be significant. The original scaling analysis relies on the fidelity of the discretization of the relevant Stochastic Differential Equation with SGD; it is unclear how well this approximation holds in our setting, when accounting for boundary effects.
>
>  In any case, the differences pointed to in Figure 8 actually seem to be relatively small in magnitude (which was our point), so we do not think this should be a cause for concern here.
>
>
> 3. The focus of this work is to probe the nature of universal perturbations, rather than view them from an attack and defense perspective (like most of the previous works on universal perturbations do). With this goal in mind, we believe that the targeted perturbations are the most natural ones for our study. In particular, they allow us to isolate features (positively) correlated with a particular target. In fact, prior work (Moosavi-Dezfooli et al.) indicates that untargeted universal perturbations generally perturb most examples towards a single common target. So this suggests that they may have similar properties to the targeted universal perturbations.
>
>  Also, note that all of the “standard” adversarial perturbations used in the experiments are also targeted; we are only varying the level of universality in their comparisons. (In Section 5.3, we go further and look at more intermediate interpolations between the two extremes.)
>
>  Our analysis of spatial invariance was motivated by the desire to illustrate the difference between universal and standard adversarial perturbations. As the reviewer suggested, one can impose this as an additional constraint when computing adversarial perturbations; this may be another way to “restrict” adversarial perturbations in order to isolate different non-robust features.

---

> > ### Comment · AnonReviewer1 · 2020-11-24
> > **Thanks for the responses**
> >
> > Thanks for the responses. My concerns are well addressed. In general, I think it is an interesting paper, and I am willing to increase my score from 6 to 7.
> >
> > As a side note, for the response regarding Q2, though "it is unclear how well this approximation holds in our setting when accounting for boundary effects", the prior work [1] empirically shows the lr scaling rule is compatible with PGD optimization (e.g., Table 6 in [1] suggests that the model robustness is not very sensitive to batch size in adversarial training, if lr scaling rule is applied). But anyway, I agree with the authors that this is just a minor issue.
> >
> > [1] Xie, Cihang, and Alan Yuille. "Intriguing properties of adversarial training at scale." ICLR 2020

---

> > > ### Author Response · Authors · 2020-11-24
> > > **Thank you**
> > >
> > > Thanks again for the constructive feedback and comments.

---

### Author Response · Authors · 2020-11-17
**Revised draft**

We thank all reviewers for their valuable feedback.

We have uploaded a revised draft with the following changes:

* Overall, we polished the writing significantly.

 In particular, we are more consistent and clear in our discussion of semantics and human-alignedness.

* We added material to the Appendices:
    * A.3. More experiments on bounding robust feature leakage to supplement results in the main paper.
    * B.1. More information about the dataset.
    * D. New discussion on l2 vs linf perturbations.

---

### Decision · Program_Chairs · 2021-01-07
**Final Decision**

**Decision:**

Reject

**Comment:**

The reviews were a bit mixed, and there was some concern on the usefulness and actual novelty of this work. On one hand, the authors did a nice job in visualizing their findings and conducting a wealth of interesting experiments. On the other hand, the submission suffers severely from hand-waving definitions and arguments. Many terms were not precisely defined, various hyperparameters were not thoroughly investigated, and yet conclusions were made based on indirect experimental results. The AC agrees with the reviewers that it is not very clear how this work would impact the field. For exploratory work like this one, there is also a great danger that one may simply overfit the observations and squeeze conclusions from thin air. It would be more convincing if the authors could largely quantify their definitions and results. For example: what do we mean by human-aligned? robust / nonrobust feature? (this definition depends on the perturbation size hence needs more elaboration.) Is there any way to quantify the results in Fig 2, including the impact of epsilon? Should these adversarial examples be called universal if their ASR falls below what threshold? Are (some of) the conclusions (e.g. translation invariance, semantic) a direct consequence of the perturbation being universal? At this stage this work would be an excellent workshop paper but a bit more rigor would be needed for publishing at the ICLR conference.